# Generalized Mathematical Model of the Grain Drying Process

**Ryszard Myhan and Marek Markowski ***

Faculty of Engineering, University of Warmia and Mazury in Olsztyn, Oczapowskiego 2, 10-719 Olsztyn, Poland
* Correspondence: marek@uwm.edu.pl; Tel.: +48-89-523-34-13

**Abstract:** Convective cereal grain drying is an energy-intensive process. Mathematical models are applied to analyze and optimize grain drying processes in different types of dryers and in different stages of drying to improve final grain quality and reduce energy consumption. The aim of the present study was to develop a generalized mathematical model of the grain drying process that accounts for all drying stages, including loading and unloading of unprocessed grain, drying, and cooling of dry grain. The developed mathematical model is a system of algebraic equations, where the calculated coefficients are determined by the thermophysical and diffusive properties of dried grain. The model was validated for batch drying of wheat, canola, and corn grain, as well as continuous flow drying of wheat grain. The results were compared with published findings. The relationships between energy consumption during drying and drying time vs. air temperature at the dryer inlet and air stream volume were determined. Dryer capacity and drying conditions specified by the manufacturers, as well as loading and unloading capacity, were considered during batch drying. Continuous flow drying simulations were conducted in counter-flow, parallel-flow, and cross-flow mode. Simulation results indicate that the proposed models correctly depicted process flow in both batch and continuous flow dryers.

**Keywords:** cereal grain; drying; mathematical model; computer simulation

## 1. Introduction

Cereal grain is a staple food worldwide. Cereal grain with low moisture content can be stored for up to several years under controlled conditions. The storage life of cereal grain is determined mainly by temperature, moisture content, and layer thickness. Batches of combine-harvested grain transported to the drying plant can differ in moisture content. Grain with high moisture content has to be dried before storage. The drying process is expensive; therefore, food producers have a preference for grain with low moisture content. However, excessive shedding of dehydrated grain during harvest can lead to grain losses [1–3]. The optimal moisture content of harvested wheat, barley, rye, and rice grain is 14–17%, but these crops can be harvested already when grain moisture content is 20–22% [4–7]. Corn grain should be harvested when its moisture content reaches 35–40% [8,9] and canola grain—when its moisture content reaches 11–12% [10,11]. Grain with 23% moisture content can be safely stored at a temperature of 35 °C for less than 20 h. When relative humidity is constant, the equilibrium moisture content (EMC) of grain decreases by around 0.5% per every 10 °C increase in air temperature [12]. Therefore, grain can be safely stored for 35 days when its moisture content is reduced to 14% and batch temperature is reduced to 25 °C [13,14].

The safe moisture content of wheat, corn, sorghum, and rice grain for long-term storage is estimated at 12%. In most cases, additional drying is required to achieve this value. Grain is dried with the use of various dryers in drying plants [15,16]. Dryers can be operated in continuous mode (continuous-flow dryers) or batch mode (batch dryers). Continuous-flow dryers can be operated in parallel-flow, counter-flow, or cross-flow mode. Various dryers that differ in structure and operating parameters are available on the market. The selection of an optimal dryer that meets a producer's specific needs and processing

capacity is difficult. Digital simulations of the grain drying process based on mathematical models can be very helpful in the decision-making process. Therefore, the aim of this study was to develop a generalized mathematical model for simulating different stages of the grain drying process, including loading and unloading of unprocessed grain, drying, and cooling of dry grain. The model should simulate dryers operating in both continuous-flow and batch mode, as well as different process configurations (parallel-flow, counter-flow, and cross-flow mode).

## 2. Generalized Mathematical Model of a Grain Dryer

The generalized model was developed in four stages. In the first stage, the drying process was decomposed into sub-processes. Grain loading and unloading operations were described in the second stage. The phenomena occurring in the drying chamber were described in the third stage. The process of cooling dried grain was described in the fourth stage.

### 2.1. Decomposition of the Drying Process

Four types of streams were identified in the model: grain dry matter stream $S_g$, impurities stream $S_i$, water stream $S_w$, and air stream $S_a$. Each stream represents the flow of matter. Streams $S_g$, $S_i$, and $S_w$ represent moist and contaminated grain. During the drying process, a part of the water stream $\Delta S_w$ is lifted with the air stream (Figure 1). All the symbols used in this paper are summarized in Nomenclature.

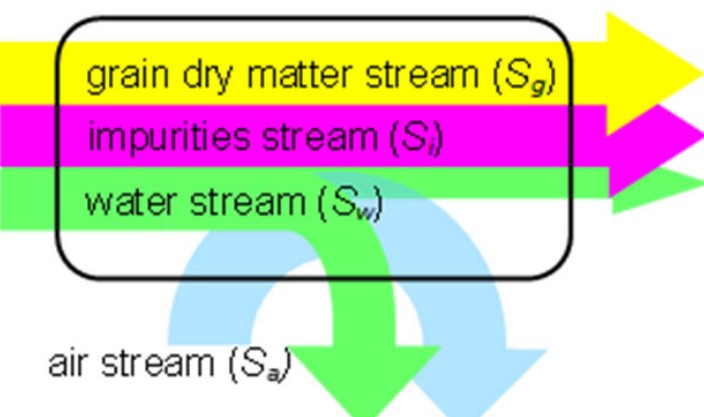

**Figure 1.** Flow and division of streams during the drying process.

All types of grain dryers, irrespective of differences in their structure, perform loading, drying, cooling, and unloading operations. In batch dryers, these operations are separated in time and conducted in a specified order. In continuous-flow dryers, grain loading and unloading operations consist of two stages. During loading, the drying chamber is filled with moist grain in the first stage, and grain is supplied continuously to the chamber in the second stage. During unloading, dried grain is regularly removed from the drying chamber, and the chamber is completely emptied in the final stage of the drying process (Figure 2).

Various mathematical models can be used to describe complex drying machines and their operations. For this reason, a grain dryer was decomposed into the following functional units: loading unit, drying chamber, cooling unit, unloading unit, and source of drying air (Figure 3). Each functional unit will be modeled in subsequent parts of the study.

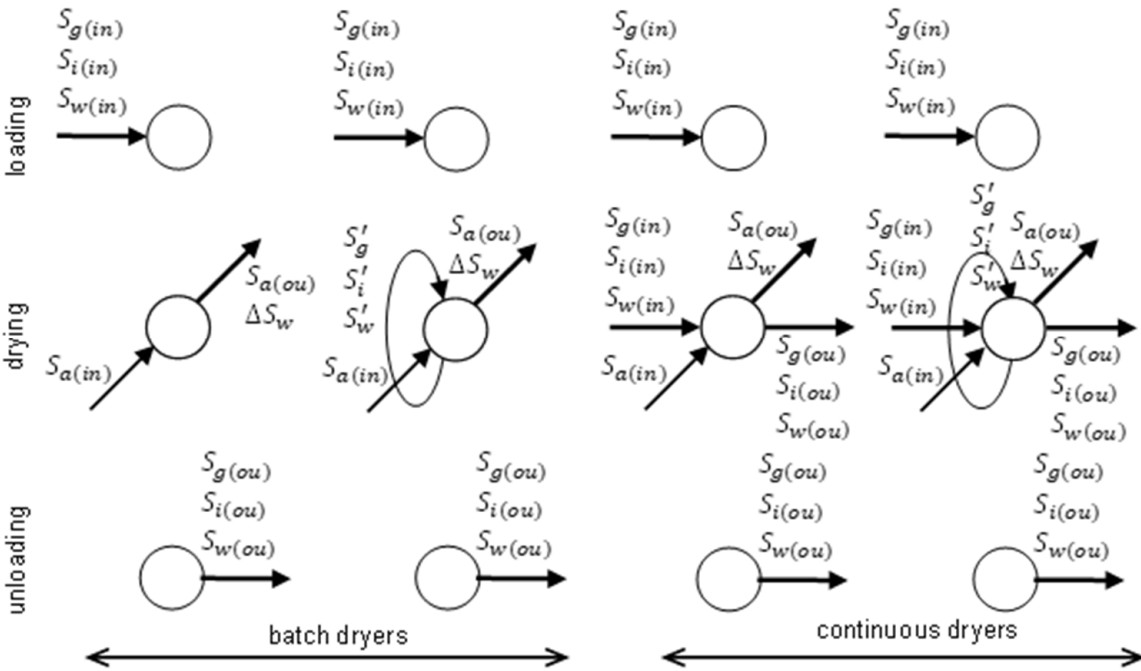

**Figure 2.** Operations performed by batch dryers and continuous-flow dryers.

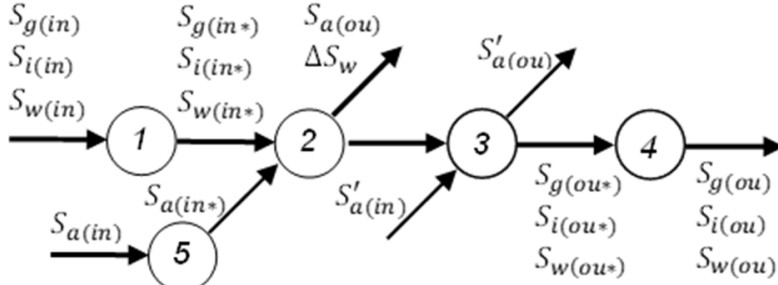

**Figure 3.** Decomposition of the grain dryer into functional units. 1. Loading unit; 2. drying chamber; 3. cooling unit; 4. unloading unit; 5. source of drying air.

### 2.2. Loading and Unloading Operations

The following parameters were considered when modeling grain loading operations: (a) grain mass at the dryer inlet (grain dry matter, impurities, water); (b) loading time; (c) grain damage; (d) heat consumption during preliminary grain drying (considered in the analysis of the dryer's energy consumption). For batch dryers and loading operations in continuous-flow dryers, the parameters described in points (a), (b), and (c) can be expressed with the use of the following equations:

$$
\begin{cases}
S_{g(in)} = S_{g(in*)} + \Delta S_{g(lo)} \\
S_{i(in)} = S_{i(in*)} - \Delta S_{g(lo)} \\
S_{w(in)} = S_{w(in*)} \\
S_{g(in)} + S_{i(in)} + S_{w(in)} \leq E_{(lo)} \\
S_{g(lo)} = a_1 + a_2 \cdot \frac{S_{w(in)}}{S_{g(in)} + S_{i(in)}} \\
t_{(lo)} = \frac{\rho_{(lo)} \cdot V_{dr}}{S_{g(in)} + S_{i(in)} + S_{w(in)}} \\
S_{g(ou)} = S_{i(ou)} = S_{w(ou)} = 0
\end{cases}
\tag{1}
$$

where (*in*∗) refers to grain dry matter, impurities, and water streams at the dryer inlet, directly behind the loading unit; $\Delta S_{g(lo)}$ refers to grain losses in the loading unit; $E_{(lo)}$ is nominal capacity of the loading unit; $t_{(lo)}$ is loading time; $\rho_{(lo)}$ is average moisture content

of moist and contaminated grain calculated with an empirical equation [17]; $V_{dr}$ is volume of the drying chamber; $a_1$ and $a_2$ are empirical coefficients of grain damage [18].

The following parameters were considered when modeling grain unloading operations in both types of dryers: (a) grain mass at the dryer inlet (grain dry matter, impurities, water); (b) unloading time; (c) grain damage in the unloading unit; (d) heat evacuated with dried grain. For batch dryers and for unloading operations in continuous-flow dryers, the above parameters can be expressed with the use of the following equations:

$$\begin{cases} S_{g(ou)} = S_{g(ou*)} - \Delta S_{g(un)} \\ S_{i(ou)} = S_{i(ou*)} + \Delta S_{g(un)} \\ S_{w(ou)} = S_{w(ou*)} \\ S_{g(ou)} + S_{i(ou)} + S_{w(ou)} \leq E_{(un)} \\ S_{g(un)} = a_1 + a_2 \cdot \frac{S_{w(ou)}}{S_{g(ou)} + S_{i(ou)}} \\ t_{(un)} = \frac{\rho_{(un)} \cdot V_{dr}}{S_{g(ou)} + S_{i(ou)} + S_{w(ou)}} \\ S_{g(in)} = S_{i(in)} = S_{w(in)} = 0 \end{cases} \tag{2}$$

where: $(ou*)$ refers to grain dry matter, impurities, and water streams at the dryer outlet, ahead of the unloading unit; $\Delta S_{g(un)}$ is grain loss in the unloading unit; $E_{(un)}$ is nominal capacity of the unloading unit; $t_{(un)}$ is unloading time; $\rho_{(un)}$ is average density of dried and contaminated grain; $a_1$, $a_2$ are empirical model coefficients.

During the supply of moist grain and the removal of dried grain from a continuous-flow dryer, the relationships between the analyzed streams can be described with the following equations (Equation (3)):

$$\begin{cases} S_{g(ou)} = S_{g(in)} - \left( \Delta S_{g(lo)} + \Delta S_{g(dr)} + \Delta S_{g(un)} \right) \\ S_{i(ou)} = S_{i(in)} + \left( \Delta S_{g(lo)} + \Delta S_{g(dr)} + \Delta S_{g(un)} \right) \\ S_{w(ou)} = S_{w(ou)} - \Delta S_w \\ S_{a(in)} = S_{a(in)} \\ S_{g(in)} + S_{i(in)} + S_{w(in)} + \left( S'_g + S'_i + S'_w \right) \leq E_{(lo)} \\ S_{g(ou)} + S_{i(ou)} + S_{w(ou)} + \left( S'_g + S'_i + S'_w \right) \leq E_{(un)} \end{cases} \tag{3}$$

where: $\Delta S_{g(dr)}$ is loss of grain dry matter during drying; $\Delta S_w$ is water stream evacuated from dried grain; $S'_g$, $S'_i$, and $S'_w$ are mixing of grain dry matter, impurities, and water streams in a continuous-flow dryer, respectively.

## 2.3. Drying Chamber

### 2.3.1. Classification of the Models

The following closely related phenomena are considered in an analysis of the drying process: heat and mass transfer in dried material and drying air; changes in the moisture content, density, and temperature of dried material and drying air; velocity and direction of grain and air flow [19,20]. Models with different degrees of generalization are applied to simulate complex physical phenomena and the mutual relationships between these processes: (a) convective drying models for solids; (b) convective drying models for grain layers. The latter category of models is used to describe grain drying processes in industrial dryers.

Convective drying models for solids are based directly on the mathematical theory of convective drying of solid materials [21,22]. This group of models includes convective drying equations for solids with a defined geometry, such as models describing water diffusion in a sphere, an infinite slab or a cylinder [21], models of heat and mass transfer in individual seeds [23], and numerical models that rely on the finite element method to describe and simulate temperature and moisture distribution inside individual kernels [24].

Convective drying models for grain layers can be regarded as an elaboration on convective drying models of solids and they include utilitarian functions. These models account for the shape and size of individual grains, and they are generally applied to describe the drying process of specific grain types. In some cases, these models also account for the unequal distribution of selected grain parameters. Distributed- and lumped-parameter models, as well as equations for modeling thin grain layers, can be applied [21]. In turn, models describing grain drying in a thick layer more accurately approximate real-world conditions. These models are based on the general theory of convective drying and they often involve empirical formulas. In these models, the considered phenomena are non-stationary processes, which implies that grain temperature $T_g$, air temperature $T_a$, moisture content of grain $u_g$, and humidity $x_a$ change over time and are influenced by the position of individual grains inside the layer [25]. Based on the direction and orientation of air and grain stream flows (Figure 4), the second group of models can be subdivided into: drying models of an immobile grain layer, parallel-flow drying models, counter-flow drying models, and cross-flow drying models. To better approximate real-world conditions, some authors have also identified mixed-flow drying models [26].

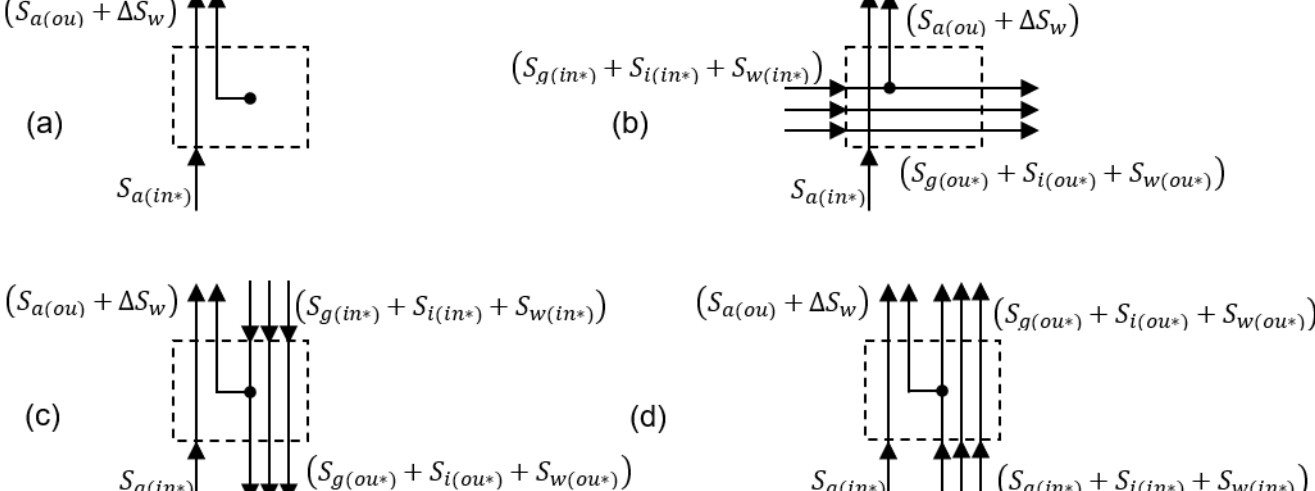

**Figure 4.** Classification of drying models. (**a**) Immobile grain layer; (**b**) cross-flow; (**c**) counter-flow; (**d**) parallel-flow.

Models describing convective drying of a grain layer involve thin-layer drying equations and equations for calculating equilibrium moisture content, enthalpy of dried grain and drying air, as well as initial and boundary conditions. However, many of these models feature simplified solutions to facilitate their mathematical formalization and applicability [27]. In situations where a quick solution to a problem is required, for example, when process parameters are automatically controlled, mathematical models are often replaced with statistical models [28,29] or neural network models [30,31]. Models belonging to the first and second group are applied to describe grain drying in various types of industrial dryers. These models are implemented in numerical simulations to predict drying results, control the drying process, and optimize dryer structure during design.

### 2.3.2. Generalized Drying Model

The above review indicates that a simple and generalized model for all types of grain dryers is very difficult to design, and the results generated by such a model can provide only approximate information about the drying process. Despite the above, an attempt to design a generalized drying model was made on the assumption that the main purpose of the modeling process at this stage is to quantify the water stream evacuated from

dried grain $\Delta S_w$. It was assumed that the amount of water removed from dried grain is affected by:

- Input variables describing impurities and moist grain streams at the drying chamber inlet $(S_{g(in*)}, S_{i(in*)}, S_{w(in*)})$;
- Process parameters that are influenced by grain type, such as kernel geometry, bulk density $\rho_{(dr)}$, equilibrium moisture content $u_r$, and effective water diffusion coefficient $D$;
- Control variables represented by the initial parameters of drying air (mass flow rate $S_{a(in*)}$, humidity $\varphi_{a(in)}$, temperature $T_{a(in)}$, average velocity $v_{a(in)}$);
- Dryer type and structural parameters;
- Average drying time in the drying chamber $t_{dr}$.

These general assumptions were based on an "iterative" drying model of an immobile grain layer and an empirical thin-layer drying equation. Layer height $h$ and drying time $t_{dr}$ were discretized (Figure 5).

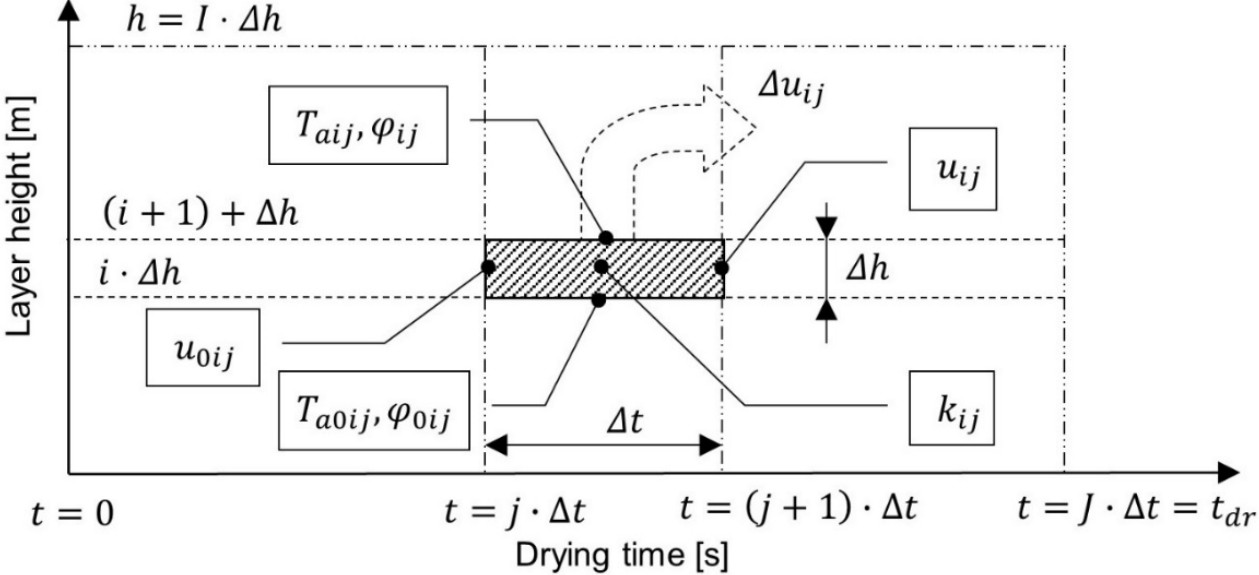

**Figure 5.** Schematic diagram of the drying model.

The total amount of water evacuated from dried grain per 1 kg of grain dry matter was calculated with the use of Formula (4):

$$\Delta U = \sum_{j=1}^{J}\left[\sum_{i=1}^{I}\Delta u_{ij}\right] = \sum_{j=1}^{J}\left[\sum_{i=1}^{I}\left(u_{0ij} - u_{ij}\right)\right], \tag{4}$$

where $u_{0ij}$ is average initial moisture content in the $i$-th layer at the beginning of the $j$-th drying period; $u_{ij}$ is average final moisture content in the $i$-th layer at the end of the $j$-th drying period.

The average amount of removed water $\Delta S_w$ was calculated with the use of Equation (5):

$$\Delta S_w = \Delta U \cdot \frac{t_{lo}\cdot\left(S_{g(in*)} + S_{i(in*)}\right)}{t_{dr}}, \tag{5}$$

where $t_{lo} = t_{dr}$ for continuous-flow dryers. The average final moisture content of grain in the $i$-th layer at the end of the $j$-th drying period was calculated with the use of the thin-layer drying Equation (6):

$$u_{ij} = \left(u_{0ij} - u_{rij}\right)\cdot A\cdot exp\left(-k_{ij}\cdot\Delta t\right) + u_{rij}, \tag{6}$$

where $k_{ij}$ is drying coefficient; $u_{rij}$ is equilibrium moisture content. Both parameters are influenced by the thermophysical properties of dried material, as well as the temperature $T_a$ and relative humidity $\varphi$ of drying air:

$$\begin{cases} k_{ij} = f\left(\overline{T}_{aij}, \overline{\varphi}_{ij}\right) \; ; \quad u_{rij} = f\left(\overline{T}_{aij}, \overline{\varphi}_{ij}\right) \\ \overline{T}_{pij} = \frac{T_{0ij}+T_{ij}}{2}; \quad\quad\quad \overline{\varphi}_{ij} = \frac{\varphi_{0ij}+\varphi_{ij}}{2} \end{cases} \tag{7}$$

where $T_{0ij}$ and $T_{ij}$ denote air temperature at the "inlet" and "outlet" of the $i$-th layer, respectively.

The drying coefficient can be calculated using the equation developed by Pabis and Henderson [32] or other equations for calculating the drying coefficient for various cereals that were proposed in the literature and reviewed by Kaleta [33]. Equilibrium moisture content and the drying coefficient can be also calculated with empirical formulas, which have been validated for various types of grain, and different relative humidity and temperature of drying air. The solutions proposed by numerous authors, including Oswin, Cheng-Pfost, Chen-Clayton, Henderson, and Nellist, were reviewed by Kaleta [33], and their applicability was verified by Chen and Jayas [34]. Thin-layer drying kinetics equations for plant materials, including cereal grain, were also described by Erbay [35].

Heat transfer in the $i$-th layer in time $\Delta t$ was calculated on the assumption that the difference between the amount of heat supplied $Q_{a(in)}$ and evacuated $Q_{a(ou)}$ with drying air was used entirely to evaporate water $Q_{r(\Delta u)}$ and increase grain temperature $\Delta Q_{g(\Delta t)}$. Heat transfer was calculated with the use of Equation (8):

$$\begin{cases} Q_{a(in)} = Q_{a(ou)} + Q_{r(\Delta u)} + \Delta Q_{g(\Delta t)} \\ Q_{a(in)} = S_{a(in*)} \cdot \frac{T_{a0ij}}{1+x_{0ij}} \cdot (c_a + x_{0ij} \cdot c_v) \cdot \Delta t \\ Q_{a(ou)} = S_{a(in*)} \cdot \frac{T_{aij}}{1+x_{0ij}} (c_a + x_{0ij} \cdot c_v) \cdot \Delta t + \frac{V_{dr} \cdot \rho_{ij}}{I} \left(u_{0ij} - u_{ij}\right) \cdot c_v \cdot T_{gij} \\ Q_{r(\Delta u)} = \frac{V_{dr} \cdot \rho_{ij}}{I} \left(u_{0ij} - u_{ij}\right) \cdot r_{ij} \\ \Delta Q_{g(\Delta t)} = \frac{V_{dr} \cdot \rho_{ij}}{I} \left(c_g + u_{ij} \cdot c_w\right) \cdot \left(T_{gij} - T_{g0ij}\right) \\ \Delta Q_{g(\Delta t)} = (a\alpha) \frac{V_{dr}}{I} \Delta t \end{cases} \tag{8}$$

where $V_{dr}$ is volume of the drying chamber; $c_g$, $c_a$, $c_w$, and $c_v$ are specific heat of grain dry matter, dry air, water, and steam, respectively; $x_{0ij}$ and $x_{ij}$ are initial and final humidity, respectively; $\rho_{ij}$ is average grain density in the $i$-th layer; $r_{ij}$ is heat of water evaporation from the $i$-th grain layer; $T_{g0ij}$ and $T_{gij}$ denote initial and final grain temperature; $(a\alpha)$ is heat transfer coefficient.

Initial and final humidity in the $i$-th layer was determined with the use of Equation (9):

$$\begin{cases} x_{0ij} = \frac{\varphi_{0ij}}{100 - \varphi_{0ij}} \\ x_{ij} = \frac{\varphi_{ij}}{100 - \varphi_{ij}} \\ x_{ij} = x_{0ij} + \frac{V_{dr} \cdot \rho_{ij} \cdot (1 + x_{0ij}) \cdot (u_{0ij} - u_{ij})}{S_{a(in*)} \cdot \Delta t \cdot I} \end{cases} . \tag{9}$$

Average grain density in the $i$-th layer was calculated with Equation (10):

$$\rho_{ij} = \rho_g \cdot \left(1 + \frac{u_{0ij} + u_{ij}}{2}\right) \tag{10}$$

Heat of water vaporization from the $i$-th layer is affected by the type of dried material, its moisture content, and temperature, and it can be determined with the use of an empirical formula proposed by Gallaher [33]:

$$r_{ij} = r \left[1 + 1.167 \cdot exp\left(-18.04 \cdot \frac{u_{0ij} + u_{ij}}{2}\right)\right] \tag{11}$$

The heat transfer coefficient can be calculated with the model proposed by Miketinac et al. [36] or Equation (12) [37]:

$$(a\alpha) = a_e \cdot 1.3 \cdot S_{a(in*)}^{0.59} \left[ \frac{0.0006 \cdot (1.72 + 0.00463 \cdot T_{aij})}{d_e} \right]^{0.41} \tag{12}$$

where $a_e$ is the ratio of seed area to seed volume; $d_e$ is the geometric mean diameter.

Regardless of the applied drying method (immobile layer, parallel-flow, counter-flow, cross-flow), the above model will be always characterized by identical initial conditions:

$$(t = 0 \wedge j = 1) \Rightarrow \forall_{i \in <1,I>} \left[ (u_{0ij} = u_0) \wedge (T_{g0ij} = T_{g0}) \right] \tag{13}$$

and boundary conditions:

$$(t \in <0, t_{dr}> \wedge i = 1) \Rightarrow \forall_{j \in <1,J>} \left[ (T_{a0ij} = T_{a0}) \wedge (\varphi_{0ij} = \varphi_0) \right] \tag{14}$$

where $u_0$ and $T_{g0}$ are initial moisture content and initial temperature of grain in the drying chamber, respectively; $T_{a0}$ and $\varphi_0$ are air temperature and humidity at the drying chamber inlet, respectively.

The algorithm for calculating the parameters of drying air at the inlet of successive grain layers remains unchanged, and it can be expressed with the following Formula (15):

$$(\forall_{j \in <1,J>} \wedge \forall_{i \in <2,I>}) \left[ (T_{a0ij} = T_{a(i-1)j}) \wedge \left( \varphi_{0ij} = \varphi_{(i-1)j} \right) \right] \tag{15}$$

In turn, the drying method affects the "iterative" algorithm for estimating the parameters of dried grain in successive layers and time steps. Regardless of the adopted time step, the algorithm for grain dried in an immobile layer can be expressed as follows:

$$(\forall_{j \in <2,J>} \wedge \forall_{i \in <1,I>}) \left[ \left( u_{0ij} = u_{i(j-1)} \right) \wedge \left( T_{g0ij} = T_{g0i(j-1)} \right) \right] \tag{16}$$

In parallel-flow and counter-flow modes, the algorithm can be significantly simplified on the assumption that the time step $\Delta t$ can be described by the following Formula (17):

$$\Delta t = \frac{V_{dr} \cdot \rho_{IJ}}{I \cdot (S_{g(ou)} + S_{i(ou)} + S_{w(ou)})} \tag{17}$$

The above implies that a single grain layer is evacuated from the drying chamber in time $\Delta t$. The first grain layer ($i = 1$) is removed in the counter-flow mode, which can be described with the below Dependency (18):

$$\begin{cases} (\forall_{j \in <2,J>} \wedge \forall_{i=I}) \left[ (u_{0ij} = u_0) \wedge (T_{g0ij} = T_{g0}) \right] \\ (\forall_{j \in <2,J>} \wedge \forall_{i \in <1,(I-1)>}) \left[ \left( u_{0ij} = u_{(i+1)(j-1)} \right) \wedge \left( T_{g0ij} = T_{g(i+1)(j-1)} \right) \right] \end{cases} \tag{18}$$

The last grain layer ($i = I$) is removed in the parallel-flow mode, and the corresponding equation is Equation (19):

$$\begin{cases} (\forall_{j \in <2,J>} \wedge \forall_{i=1}) \left[ (u_{0ij} = u_0) \wedge (T_{g0ij} = T_{g0}) \right] \\ (\forall_{j \in <2,J>} \wedge \forall_{i \in <2,I>}) \left[ \left( u_{0ij} = u_{(i-1)(j-1)} \right) \wedge \left( T_{g0ij} = T_{g(i-1)(j-1)} \right) \right] \end{cases} \tag{19}$$

Continuous-flow dryers have been designed for processing the grain of all cereal species, corn, oilseed crops, and legumes, which is directly consumed, processed into feed, or used for sowing. During the drying process, grain is circulated in tower dryers until the required moisture content (set indirectly) is achieved, regardless of its initial moisture content. Continuous grain flow and grain mixing in the drying chamber will lead to changes in the presented equations and algorithms. In a batch dryer, Equation (16)

should be replaced with Formulas (17) and (18), and the following modifications should be introduced:

$$
\begin{cases}
\Delta t = \dfrac{V_{dr} \cdot \rho_{IJ}}{I \cdot \left(S'_g + S'_i + S'_w\right)} \\[2mm]
\left(\forall_{j \in <2,J>} \wedge \forall_{i=I}\right) \rightleftarrows \left[\left(u_{0ij} = u_{1(j-1)}\right) \wedge \left(T_{g0ij} = T_{g1(j-1)}\right)\right]
\end{cases}
\tag{20}
$$

In dryers operating in counter-flow mode (Equation (18)) $u_0$ and $T_{g0}$ should be substituted with the following expressions:

$$
\begin{cases}
u_0 \to u'_0 = \dfrac{S_{w(in)} + S'_w}{\left(S_{g(in)} + S'_g + S_{i(in)} + S_{i(in)}\right)} \\[3mm]
T_{g0} \to T'_{g0} = \dfrac{T_{g0} \cdot \left(S_{g(in)} + S_{i(in)} + S_{w(in)}\right) + T_{g1(j-1)}\left(S'_g + S'_i + S'_w\right)}{\left(S_{g(in)} + S'_g + S_{i(in)} + S'_i + S_{w(in)} + S'_w\right)}
\end{cases}
\tag{21}
$$

In dryers operating in parallel-flow mode (Equation (19)), the corresponding substitutions are:

$$
\begin{cases}
u_0 \to u'_0 = \dfrac{S_{w(in)} + S'_w}{\left(S_{g(in)} + S'_g + S_{i(in)} + S_{i(in)}\right)} \\[3mm]
T_{g0} \to T'_{g0} = \dfrac{T_{g0} \cdot \left(S_{g(in)} + S_{i(in)} + S_{w(in)}\right) + T_{gI(j-1)}\left(S'_g + S'_i + S'_w\right)}{\left(S_{g(in)} + S'_g + S_{i(in)} + S'_i + S_{w(in)} + S'_w\right)}
\end{cases}
\tag{22}
$$

The drying algorithm for the cross-flow mode is also based on the proposed iterative model and Equation (17), based on the assumption that the volume of grain evacuated from the drying chamber in time $\Delta t$ is equivalent to the volume of a single grain layer. However, in this case, grain volume is divided equally by all layers, and moist grain with parameters $u_0$ and $T_{g0}$ is supplied in successive iterations. Therefore, the initial parameters of the *i*-th layer are average values. The above implies that the model will not provide information about moisture content distribution, but it will estimate the average moisture content at the dryer outlet. The drying algorithm for the cross-flow mode can be expressed as follows:

$$
\left(\forall_{j \in <2,J>} \wedge \forall_{i \in <1,I>}\right)\left[\left(u_{0ij} = \overline{u_{0ij}}\right) \wedge \left(T_{g0ij} = \overline{T_{g0ij}}\right)\right]
\tag{23}
$$

where:

$$
\begin{cases}
\overline{u_{0ij}} = \dfrac{(I-1) \cdot u_{i(j-1)} + u_0}{I} \\[3mm]
\overline{T_{g0ij}} = \dfrac{(I-1) \cdot T_{gi(j-1)} \cdot \rho_{i\ (j-1)} + T_{g0} \cdot \rho_0}{(I-1) \cdot \rho_{i\ (j-1)} + \rho_0}
\end{cases}
\tag{24}
$$

Simulations of grain damage during drying constitute a separate problem. Grain damage does not directly affect the drying process (both damaged and undamaged grain is dried), but it can significantly affect the quality of the end product by increasing the impurities stream at the expense of the grain stream (Equation (3)). Overheating is the main cause of grain damage and it can lead to: (a) decrease in germination capacity [38]; (b) changes in grain hardness [39]; (c) changes in the chemical and functional properties of proteins [40]; (d) damage to the seed coat and grain cracking [41]. These phenomena are interrelated and they cannot be examined separately. Particles that constitute impurities are classified based on the designation of dried grain (for example, grain with a low germination capacity can be used directly for consumption or processed into feed). In addition to drying temperature and time, the severity of grain damage is also determined by grain type, moisture content, dryer's design, and drying method. Grain losses resulting from overheating were determined with the use of the equation proposed by McFarlane [42] on the assumption that:

$$
\frac{\Delta S_{d(dr)}}{S_{g(in*)}} = a_1 \cdot exp\left[a_2 \cdot max\left(T_{gij}\right) + a_3 \cdot ln(u_0) + a_4\right]
\tag{25}
$$

where $max\left(T_{gij}\right)$ is maximum grain temperature; $a_1$, $a_2$, $a_3$, $a_4$ are empirically defined coefficients.

For example, the following values were calculated for wheat grain dried in a continuous-flow dryer: $a_1 = 0.0064$, $a_2 = 0.08$, $a_3 = 3.36$, $a_4 = 0.81$ [43].

*2.4. Grain Cooling*

The temperature of dried grain has to be reduced when the drying process is complete. Grain can be cooled in both batch and continuous-flow dryers. In batch dryers, grain is cooled in the drying chamber in a stream of cold air (with ambient temperature). In continuous-flow dryers, grain is cooled in a dedicated chamber. In some continuous-flow dryers, the cooling chamber can be enlarged at the expense of the drying chamber, depending on the type of processed grain.

Regardless of cooling air parameters, humidity is partly reduced during cooling [44]. Therefore, under certain conditions, the cooling process can be regarded as a drying process [45] and it can be described using the previously proposed model with some modifications. The following modifications are introduced:

(a)  The parameters of the drying air stream $S_{a(in*)}$ are replaced with the parameters of the cooling air stream $S'_{a(in)}$ (Figure 3) with the use of the below Formula (26):

$$\begin{cases} S_{a(in*)} \to S'_{a(in)} \\ \varphi_a \to \varphi'_a \\ T_a \to T'_a \end{cases} \tag{26}$$

(b)  An empirical formula for calculating the drying coefficient and equilibrium moisture content [46] was selected for the cooling process based on the thin-layer drying equation (Equation (6)).

(c)  The heat transfer coefficient was calculated with the use of Equation (10) by considering the parameters of cooling air and the direction of heat flow during grain cooling;

(d)  The initial conditions described by Equation (13) were modified as follows:

●  in a batch dryer:

$$\begin{cases} (t' = 0 \wedge j' = 1) \Rightarrow \forall_{i' \in <1, I'>}\left[\left(u'_{0i'j'} = u_{iJ}\right) \wedge \left(T'_{g0i'j'} = T_{giJ}\right)\right] \\ (I' = I) \wedge \left(J' = J \cdot \frac{t_{co}}{t_{dr}}\right) \end{cases} \tag{27}$$

●  in a continuous-flow dryer:

$$\begin{cases} (t' = 0 \wedge j' = 1) \Rightarrow \forall_{i' \in <1, I'>}\left[\left(u'_{0i'j'} = u_{IJ}\right) \wedge \left(T'_{g0i'j'} = T_{gIJ}\right)\right] \\ \left(I' = I \cdot \frac{V_{co}}{V_{dr}}\right) \wedge (t_{co} = t_{dr}) \wedge (J' = J) \end{cases} \tag{28}$$

where $V_{co}$ is the volume of the dryer's cooling compartment; $t_{co}$ is average cooling time.

(e)  The boundary conditions described by Equation (14) were modified as follows:

$$(t' \in <0, t_{cht}> \wedge i' = 1) \Rightarrow \forall_{j' \in <1, J'>}\left[\left(T'_{a0i'j'} = T'_{a0}\right) \wedge \left(\varphi'_{0i'j'} = \varphi'_0\right)\right] \tag{29}$$

(f)  In "iterative" algorithms for the first layer of cooled grain, the initial parameters $u_0$ and $T_{g0}$ were replaced with parameters $u'_0$ and $T'_{g0}$ of the grain layer evacuated from the drying chamber in successive time steps (Equations (18)–(24)):

$$\begin{cases} u_0 \to u'_0 = u_{IJ} \\ T_{g0} \to T'_{g0} = T_{gIJ} \end{cases} \tag{30}$$



### 2.5. Drying Air Source

The source of the drying air consists of two elements: a fan and an air heater (heater). The fan supplies the required amount of air to the drying chamber and the heater supplies heat that is required to achieve the desired temperature in the drying chamber. Grain dryers are equipped with fans that both supply and remove air. Fans that supply air are positioned on the side of the air inlet in the drying chamber and are often integrated with heaters. In turn, fans that remove air are positioned on the side of the air outlet and they decrease pressure inside the drying chamber. Heaters have to conform to the following energy efficiency requirements:

$$\eta_u \geq 84 + 2log(P_n) \tag{31}$$

where $\eta_u$ is energy conversion efficiency in %; $P_n$ is nominal power in kW.

In a steady state, the amount of air supplied to the drying chamber $S_{a(in*)}$ (Figure 3) can be determined empirically based on two parameters specified by the manufacturer: fan specifications and the hydraulic resistance of a grain layer through which air is passed [47]. Hydraulic resistance is determined by the dryer's structure, grain filling level, grain type, and its moisture content [48–50]. Air temperature at the heater outlet was approximated (a heater's performance is rarely specified by the manufacturer) from the heat balance Equation (32):

$$T_{a0} = T_{atm} + \frac{\eta_u \cdot P_n \cdot (1 + x_0)}{100 \cdot S_{a(in*)} \cdot (c_p + c_v)} \tag{32}$$

where $T_{atm}$ is temperature of ambient air in K.

## 3. Results and Discussion

### 3.1. Butch Dryer Model

The applicability of the batch dryer model was evaluated by examining the following parameters: convergence criteria for the numerical implementation of the model, accuracy with which the analyzed devices were modeled, accuracy with which the drying process was modeled, and the model's applicability for process optimization. All simulations were performed with fixed values of environmental parameters (air temperature—15 °C, humidity—60%, atmospheric pressure—1013.25 hPa) and a fixed time step of 10 s.

To evaluate the convergence of the numerical implementation of the model, the drying process of wheat, canola, and corn grain was simulated for typical values of initial and final moisture content, grain mass, drying temperature, and apparent air velocity (refer to the legend in Figure 6). The technical specifications of 26 batch dryers supplied by seven manufacturers were analyzed for this purpose. In each case, drying time was determined by modifying the number of grain layers from 5 to 30. The results (Figure 6) indicate that the proposed model is convergent in all cases when grain is divided into 20 layers, and significant differences in drying time were not observed when grain was divided into more than 20 layers.

The model's applicability was tested based on drying time or energy consumption, where the temperature and volume of the drying air stream at the dryer inlet are the decision variables. The initial simulation involved 10 tons of wheat grain with an initial moisture content of 20% that was dried to an average moisture content of 14%. During the simulation, air temperature was increased from 55 °C to 90 °C in steps of 5 °C, and the volume of the air stream was decreased from 0.3 $m^3s^{-1}$ to 1.9 $m^3s^{-1}$ in steps of 0.1 $m^3s^{-1}$. The results and the relationships between energy consumption and drying time vs. air temperature at the drying chamber inlet and air stream volume are presented in Figure 7.

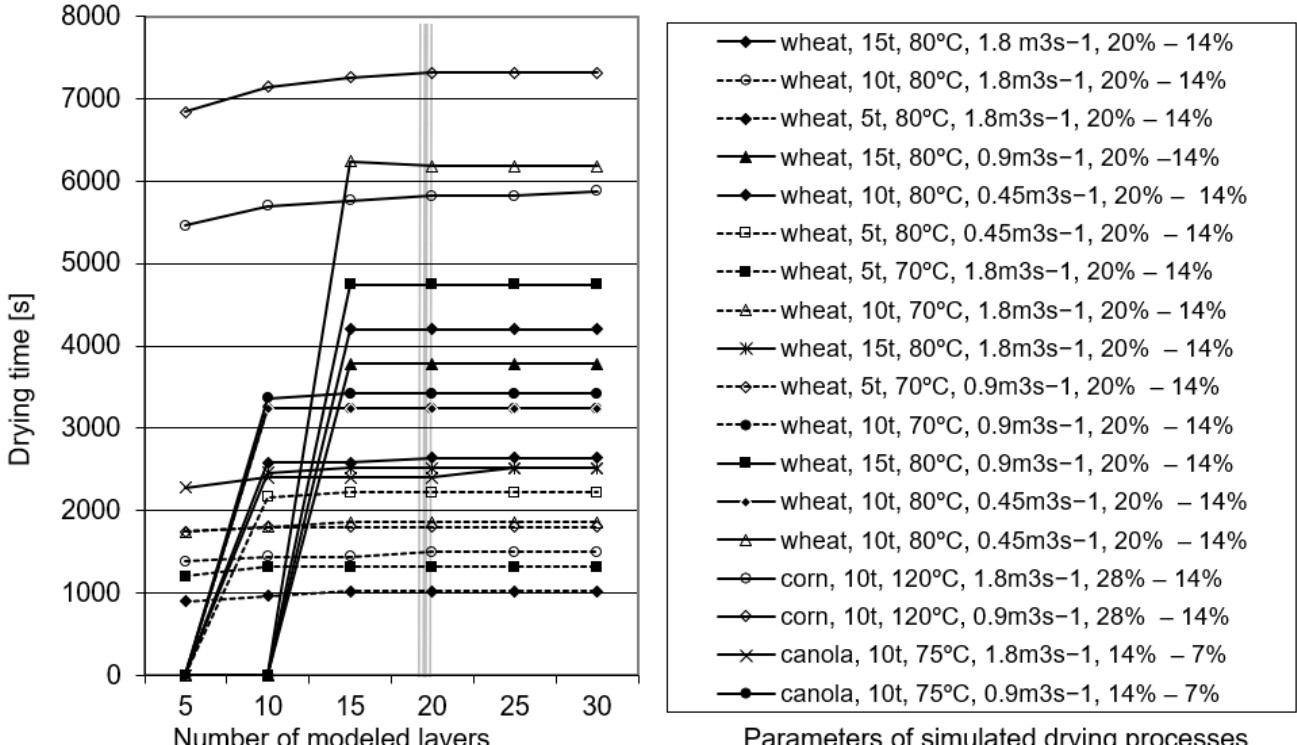

**Figure 6.** The influence of the number of layers on the drying time of wheat, canola, and corn grain.

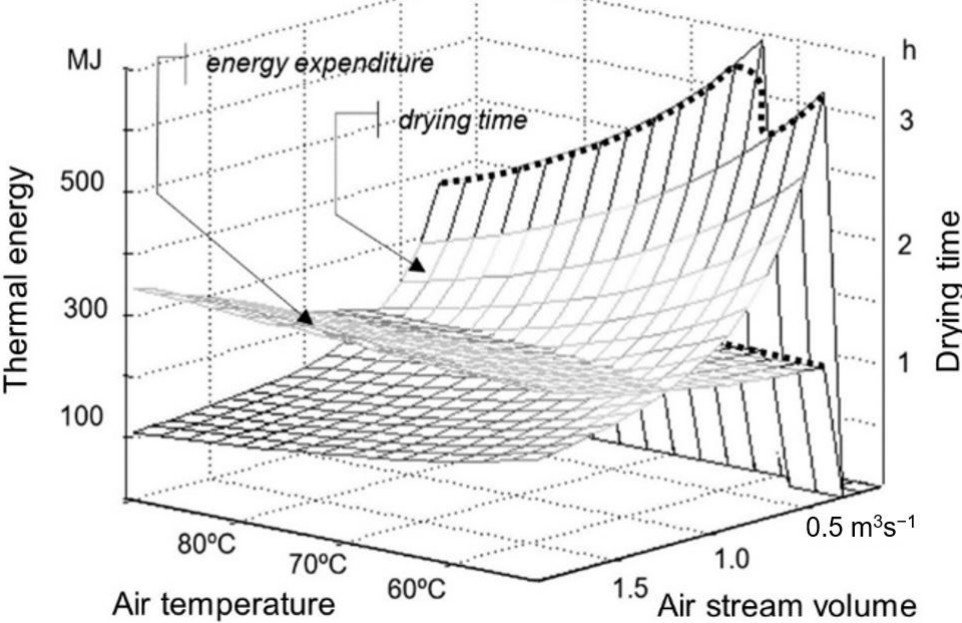

**Figure 7.** The influence of air temperature and air stream volume on energy consumption and drying time.

As expected, drying time decreased with a rise in temperature and air stream volume, whereas an increase in temperature induced a minor decrease in energy consumption, and a decrease in air stream volume induced a significant decrease in energy consumption. Energy consumption was minimized and drying time was maximized for values of the decision variable at which the relative humidity of air evacuated from the grain layer approximated 100%. In the analyzed scenario, the relative humidity of evacuated air approximated 100% when air stream volume was below 0.5 $m^3s^{-1}$ and tempera-

ture was 55–65 °C, when air stream volume was below 0.6 $m^3s^{-1}$ and temperature was 70–75 °C, and when air stream volume was below 0.7 $m^3s^{-1}$ and temperature was 80–90 °C. The above results account only for heat consumption, but not the energy consumed by the supply fan; therefore, optimal solutions can be expected in an area where both parameters interact.

The accuracy with which the analyzed devices were modeled was determined by comparing the daily capacity of real-world batch dryers with the simulated values. Twenty-six batch dryers supplied by seven manufacturers were used to simulate the drying process of wheat, canola, and corn grain. Dryer capacity, drying conditions, and loading and unloading capacity specified by the manufacturer were considered in each case. Most manufacturers do not specify air stream volume; therefore, this parameter was optimized by maximizing the dryer's performance, expressed as the ratio of product value to energy consumption. The results are presented in Figure 8. The simulated capacity for wheat, canola, and corn grain closely matched empirical results.

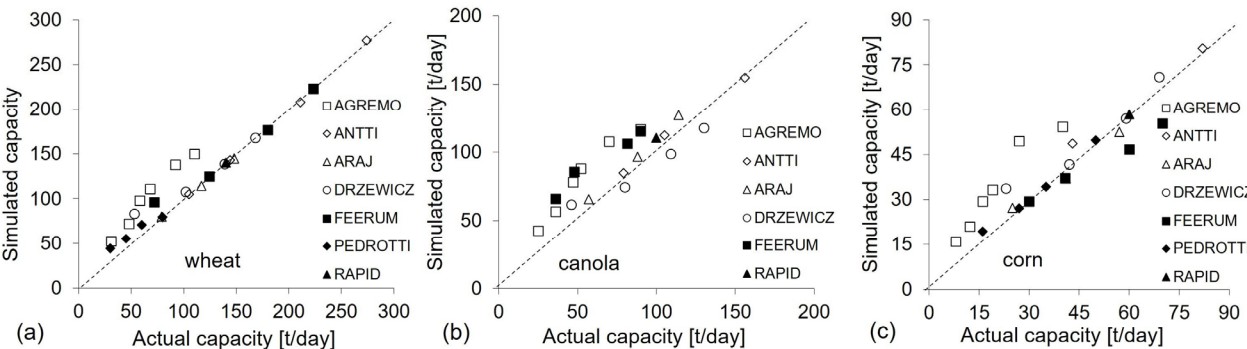

**Figure 8.** A comparison of the simulated daily capacity of selected batch dryers with their actual capacity. (**a**) Wheat; (**b**) canola; (**c**) corn.

The accuracy with which the drying process was modeled was examined by simulating changes in the moisture content of grain in successive layers relative to drying time. The drying process of 11.2 tons of wheat grain with an initial moisture content of 19% at a temperature of 80 °C (identical to real-world conditions) and minimum air velocity (when the humidity of air evacuated from the grain layer approximates 100%) is presented in Figure 9. The presented results are similar to published findings [30,51–54]. Other researchers have also reported a visible shift in the drying front [55–57]. These observations indicate that the proposed batch dryer model accurately describes the grain drying process.

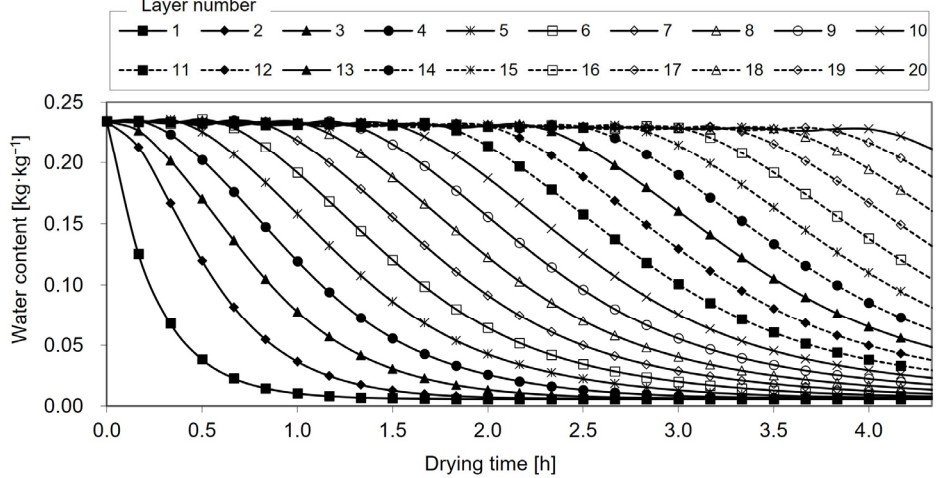

**Figure 9.** Simulated changes in the moisture content of successive layers of dried wheat grain.

### 3.2. Continuous-flow dryer Models

The grain drying process was also modeled in continuous-flow dryers. In the examples presented below, wheat grain with an initial moisture content of 19% was dried to a maximum 15% moisture content at a temperature of 80 °C. The drying process was simulated in dryers operating in counter-flow (Figure 10), parallel-flow (Figure 11), and cross-flow modes (Figure 12). The simulated changes in the moisture content of wheat grain dried in the counter-flow mode are presented in Figure 10. The results indicate that the drying process is always stable in the counter-flow mode, regardless of grain layer thickness, which confirms the model's applicability for simulation tests. Grain layers are evacuated from the drying chamber one by one (layers 5, 10, 15, and 20 are marked with a blue line in Figure 10) when the desired moisture content is achieved (blurred grey line in Figure 10) and subsequent layers are introduced to the chamber (layers 20 + 5, 20 + 10, and 20 + 15 are marked with a green line in Figure 10). The introduced grain is initially moistened and then it is gradually dried in the chamber.

Wheat grain drying in parallel-flow mode is simulated in Figure 11. In this mode, the stability of the drying process is determined by grain layer thickness. In the variant presented in Figure 11a, the drying process was modeled in a layer that was twice as thick as the layers shown in Figure 11b,c. When the layer of dried grain is very thick, grain is not evacuated continuously and uniformly from the drying chamber. Instead, grain is removed from the chamber in cycles and each cycle begins when the last layer has achieved the desired moisture content. Another disadvantage is that this drying method prolongs drying time (from the moment moist grain is loaded into the chamber and the moment when the last layer achieves the desired moisture content), and it leads to overdrying of the remaining layers and considerable differences in the moisture content of grain leaving the drying chamber. However, these differences are eliminated after a full exchange cycle has been completed and the dryer's capacity and the moisture content of evacuated grain are stabilized (Figure 11b). Preliminary drying time can be shortened and the risk of overdrying can be minimized by filling the chamber with pre-dried grain and switching to the counter-flow mode or the mixed-flow mode. A grain drying process conducted in 20% in mixed-flow mode is simulated in Figure 11c.

Wheat grain drying in cross-flow mode is modeled in Figure 12. In this mode, the moisture content of the grain layer situated at the chamber inlet is similar to the moisture content of moist grain, whereas the moisture content of evacuated grain varies considerably, from overdried (on the side of the air inlet) to underdried (on the side of the air outlet). Similar observations were made by other authors [44]. The results generated by the proposed models and the character of the observed phenomena corroborate the findings of other authors [28,30,52,54,56,58–60].

To evaluate the applicability of the proposed drying models, the simulated drying capacity was also compared with the performance of 31 real-world continuous-flow dryers supplied by three different manufacturers (ARAJ—Kąty Wrocławskie, Polska, DRZEWICZ—Nowy Drzewicz, Poland, and AG-PROJEKT—Pietrzykowice, Poland). Depending on the type of dryer, the drying process was analyzed in counter-flow mode or mixed-flow mode, where 1/3 of the grain mass was dried in the parallel-flow mode, 1/3 in the counter-flow mode, and 1/3 in the cross-flow mode. Depending on the manufacturers' specifications, the moisture content of wheat grain was reduced from 19% to 15%, or from 18% to 14%, at a temperature of 95 °C, and the remaining parameters specified by the manufacturers (grain type, tower dimensions, loading capacity, heater capacity, and heater power) were unchanged. Simulated daily capacities are compared with the manufacturers' specifications in Figure 13. The results indicate that the simulated capacities are highly similar to the actual capacities of the analyzed grain dryers. The mean relative error of the estimation of the analyzed dryers' daily capacity was 4.2%, and it did not exceed 1.5% when two dryers with the lowest capacity were excluded from the comparison.

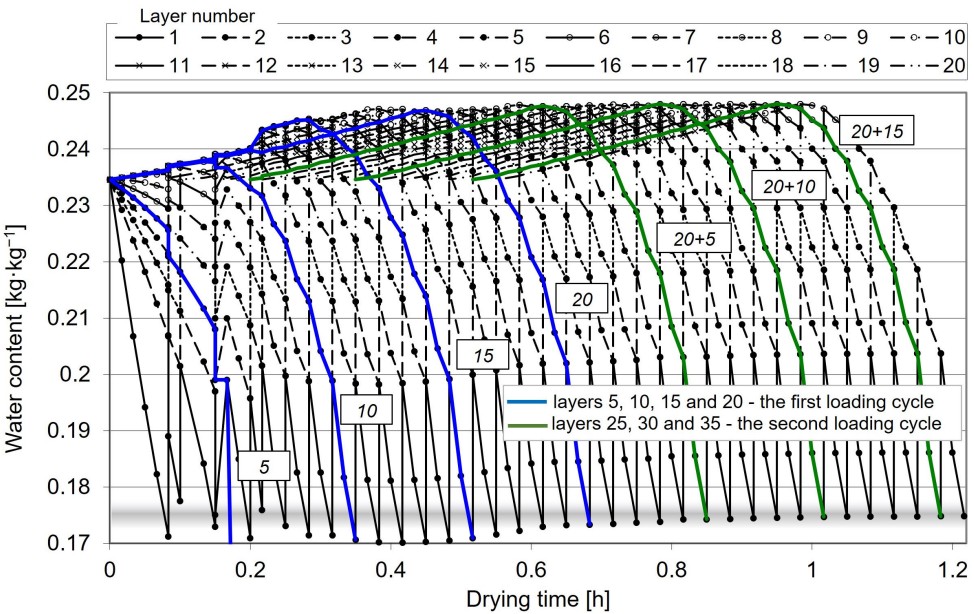

**Figure 10.** Simulated changes in the moisture content of successive grain layers dried in the counter-flow mode.

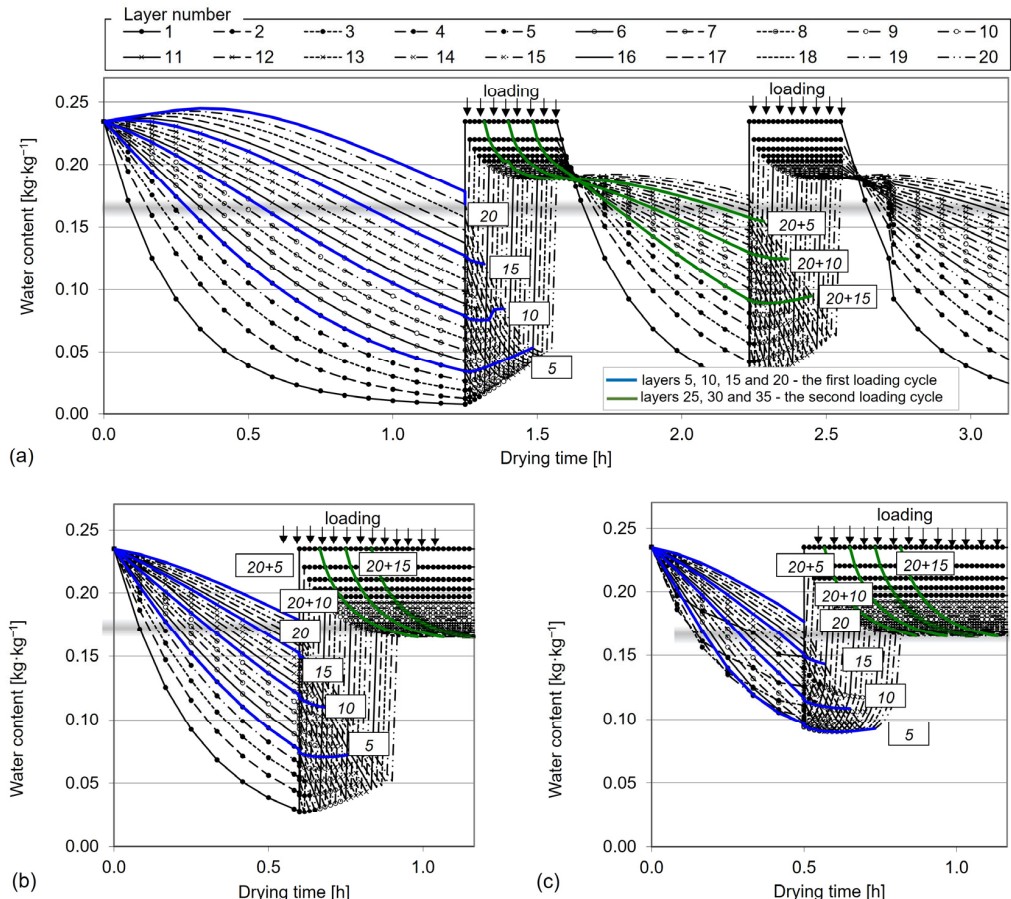

**Figure 11.** Simulation of changes in the moisture content of successive grain layers dried in the parallel-flow mode. (**a**) Cyclic drying of grain in a very thick layer; (**b**) overdrying in the initial drying period; (**c**) simulation of a drying process conducted in 20% in the mixed-flow mode.

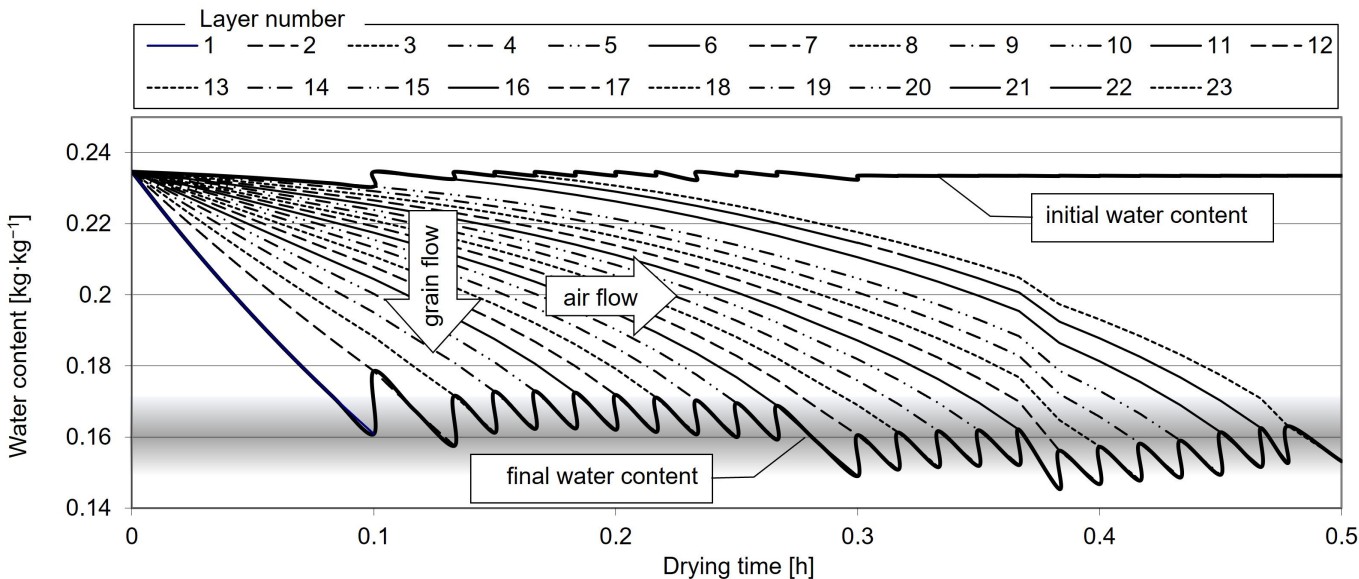

**Figure 12.** Simulated changes in the moisture content of successive grain layers dried in the cross-flow mode.

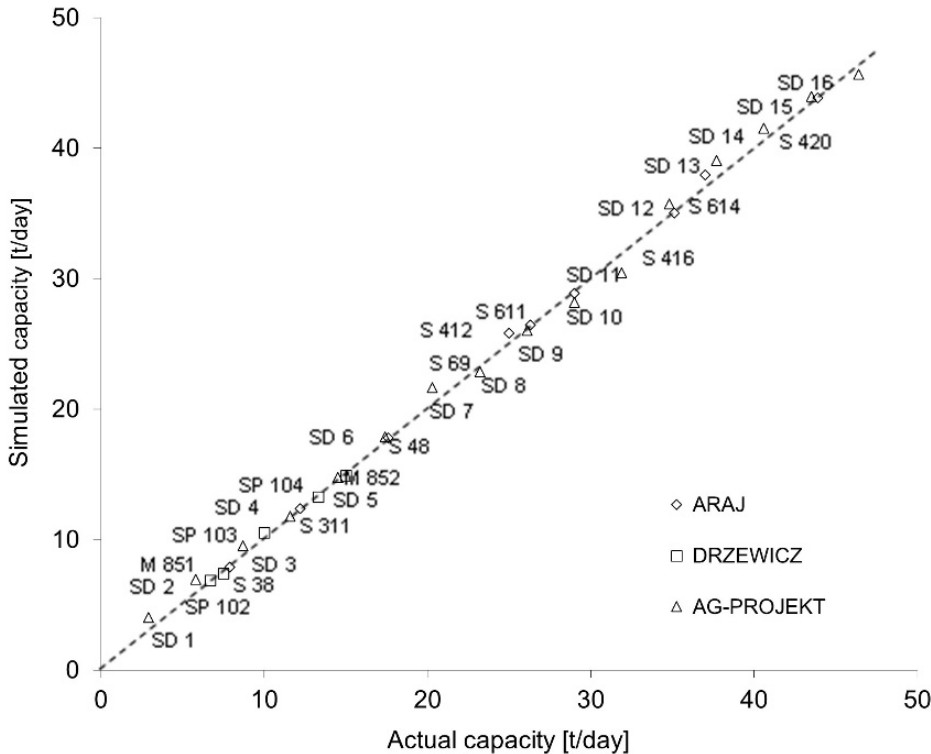

**Figure 13.** A comparison of the simulated daily capacities of selected continuous-flow dryers and the capacities specified by the manufacturers.

## 4. Conclusions

Cereal grain is a staple food worldwide. Harvested grain has to be dried before storage. Convective drying is highly energy-intensive and this process should be optimized to reduce costs.

The aim of this study was to develop a generalized mathematical model for simulating different stages of the grain drying process, including loading and unloading of unprocessed grain, drying, and cooling of dried grain. The developed model was applied to

simulate the operation of both batch dryers and continuous-flow dryers in different modes (parallel-flow, counter-flow, and cross-flow mode).

Mathematical models were developed to simulate different stages of the drying process. The proposed model is a system of algebraic equations, where the calculated coefficients are determined by the thermophysical and diffusive properties of dried grain. The model was validated for batch drying of wheat, canola, and corn grain, and for continuous-flow drying of wheat grain. Simulation results were compared with published findings. The proposed models accurately described the grain drying process in both batch and continuous-flow dryers.

The use of mathematical models to analyze and optimize convective drying of grain in various types of dryers and in different stages of the drying process can improve the quality of dried grain and minimize energy consumption. Therefore, the developed model can be used by businesses that design and build grain dryers and complete drying systems.

**Author Contributions:** Conceptualization, R.M.; Data curation, R.M.; Formal analysis, R.M. and M.M.; Investigation, R.M.; Methodology, R.M. and M.M.; Validation, R.M.; Writing—original draft, R.M.; Writing—review and editing, R.M. and M.M. Substantive revision of the work: M.M. All authors have read and agreed to the published version of the manuscript.

**Funding:** This research was financially supported by the Polish Ministry of Education and Science, and the University of Warmia and Mazury in Olsztyn, Poland [grant No. 16.601.001-300].

**Data Availability Statement:** Not applicable.

**Conflicts of Interest:** The authors declare no conflict of interest.

### Nomenclature

Mathematical symbols, subscripts, and superscripts used in this study.

| Mathematical Symbol | Description |
| --- | --- |
| $(a\alpha)$ | heat transfer coefficient in $W \cdot m^{-3} \cdot K^{-1}$ |
| $a_1$, $a_2$ | grain damage coefficients |
| $c$ | specific heat in $J \cdot kg^{-1} \cdot K$ |
| $h$ | height of the grain layer in m |
| $k$ | drying coefficient in $s^{-1}$ |
| $r$ | heat of water vaporization from grain in $J \cdot kg^{-1}$ |
| $t$ | time in s |
| $u$ | moisture content of grain dry basis in $kg \cdot kg^{-1}$ |
| $x$ | moisture content of grain wet basis in $kg \cdot kg^{-1}$ |
| $D$ | effective water diffusion coefficient in $m^2 \cdot s^{-1}$ |
| $E$ | nominal capacity in $kg \cdot s^{-1}$ |
| $S$ | mass flow stream in $kg \cdot s^{-1}$ |
| $V$ | volume of a dryer's functional unit in $m^3$ |
| $T$ | air temperature in K |
| $Q$ | heat stream in $W \cdot m^{-2}$, |
| $\Delta S_g$ | grain loss stream in $kg \cdot s^{-1}$ |
| $\Delta S_w$ | stream of evacuated water in $kg \cdot s^{-1}$ |
| $\Delta U$ | mass of evacuated water in $kg \cdot kg^{-1} \cdot s^{-1}$ |
| $\rho$ | density in $kg \cdot m^{-3}$ |
| $\varphi$ | relative humidity in % |
| $v$ | velocity in $m \cdot s^{-1}$ |

| Subscripts | |
|---|---|
| *a* | Air |
| *dr* | drying chamber |
| *g* | grain dry matter |
| *i* | impurities (material other than grain, MOG) |
| *in* | dryer inlet |
| *in∗* | inlet of a dryer's functional unit |
| *lo∗* | loading unit |
| *ou* | dryer outlet |
| *ou∗* | outlet of a dryer's functional unit |
| *un∗* | unloading unit |
| *w* | Water |

| Superscripts | |
|---|---|
| ' | stream mixing in a continuous flow dryer |

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
