# Peer review of "Generalized Mathematical Model of the Grain Drying Process"

_processes, doi:10.3390/pr10122749_

Round 1

Reviewer 1 Report

The manuscript is lacking innovation and away from originality, such that similar studies can be found in the literature. 

Author Response

Dear Editor,

Below please find our responses to the reviewers’ comments.

All revisions are marked in yellow in the manuscript.

REVIEWER #1

Comment #1:

The manuscript is lacking innovation and away from originality, such that similar studies can be found in the literature.

Answer:

The manuscript presents a new approach to developing a generalized mathematical model of the grain drying process that takes into account all stages of drying, including loading and unloading of unprocessed grain, drying and cooling of dry grain. To the best of our knowledge, such studies are scarce and our paper provides new insights to the existing knowledge base by adopting a comprehensive approach to the problem of modeling grain drying.

We are grateful to the reviewers for taking time to review our manuscript and for their valuable comments.

Yours sincerely,

Marek Markowski

Reviewer 2 Report

The article is well structured and appropriately written. 

line 215-219 : This paragraph is a duplicate sentence should be removed.

Author Response

Dear Editor,

Below please find our responses to the reviewers’ comments.

All revisions are marked in yellow in the manuscript.

REVIEWER #2

Comment #1:

Line 215-219 : This paragraph is a duplicate sentence should be removed.

Answer:

The duplicate sentence was removed.

 Comment #2:

The sentence in line 372: “There are two sources of drying air: a fan and a heater.” was marked in red.

Answer:

The sentence was changed to: "The source of the drying air consists of two elements: a fan and an air heater (heater)." The original sentence implied that there were two sources of drying air, which was not true.

We are grateful to the reviewers for taking time to review our manuscript and for their valuable comments.

Yours sincerely,

Marek Markowski

Reviewer 3 Report

The article 'A Generalized mathematical model of the grain drying process' represents an interesting study as an excellent strategy for developing a generalized mathematical model of the grain drying process that accounts for all drying stages, including loading and unloading of unprocessed grain, drying, and cooling of dry grain

An adequate number of clear figures and tables were given,

The literature review is also fine, no plagiarism issues

English can be improved.

Author Response

Dear Editor,

Below please find our responses to the reviewers’ comments.

All revisions are marked in yellow in the manuscript.

REVIEWER #3

Comment #1:

English can be improved.

Answer:

Language and style were improved, and the manuscript was revised by a native speaker of English.

We are grateful to the reviewers for taking time to review our manuscript and for their valuable comments.

Yours sincerely,

Marek Markowski